# Hints-In-Browser: Benchmarking Language Models for Programming Feedback Generation

**Nachiket Kotalwar**
MPI-SWS
nkotalwa@mpi-sws.org

**Alkis Gotovos**
MPI-SWS
agkotovo@mpi-sws.org

**Adish Singla**
MPI-SWS
adishs@mpi-sws.org

## Abstract

Generative AI and large language models hold great promise in enhancing programming education by generating individualized feedback and hints for learners. Recent works have primarily focused on improving the quality of generated feedback to achieve human tutors' quality. While quality is an important performance criterion, it is not the only criterion to optimize for real-world educational deployments. In this paper, we benchmark language models for programming feedback generation across several performance criteria, including quality, cost, time, and data privacy. The key idea is to leverage recent advances in the new paradigm of in-browser inference that allow running these models directly in the browser, thereby providing direct benefits across cost and data privacy. To boost the feedback quality of small models compatible with in-browser inference engines, we develop a fine-tuning pipeline based on GPT-4 generated synthetic data. We showcase the efficacy of fine-tuned Llama3-8B and Phi3-3.8B 4-bit quantized models using WebLLM's in-browser inference engine on three different Python programming datasets. We also release the full implementation along with a web app and datasets to facilitate further research on in-browser language models.

## 1 Introduction

Generative AI and large language models have the potential to drastically improve the landscape of computing and programming education by empowering next-generation tutoring systems. In particular, this potential lies in the advanced capabilities of state-of-the-art models like OpenAI's GPT-4 [1] to automatically generate high-quality programming feedback and provide 24/7 digital assistance to novice learners [2, 3, 4]. A series of recent works have studied these models for programming feedback generation, including benchmarking the models' performance with human tutors in terms of generated feedback quality [5], developing techniques to further improve the generation quality [6, 7, 8, 9], and deploying feedback generation technologies in classrooms to measure pedagogical benefits and students' perception of quality [10, 11, 12].

While quality is an important performance criterion, it is not the only criterion to optimize in real-world educational deployments. Let us look at a typical workflow of existing (pilot) applications that employ programming feedback techniques (see Figure 1a) [10, 11, 12]: (1) an educator deploys their technique on a server, which then communicates with state-of-the-art models like OpenAI's GPT-4 [1], and then exposes this technique to learners via an app; (2) when a learner requests a hint, metadata such as the buggy program and the learner's specific question is sent to the server to be processed by the technique; (3) once the technique generates the hint, the learner receives it back via the app. Three key issues emerge with this workflow: (i) the running costs for an educator could be 0.01–0.1 USD for a single hint request [9, 12], making it infeasible to provide large-scale free access to such technology; (ii) the waiting times for a learner could be over 30 seconds making the experience and feedback loop less interactive [9]; (iii) data privacy concerns creep in as the learner's

38th Conference on Neural Information Processing Systems (NeurIPS 2024) Track on Datasets and Benchmarks.

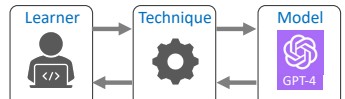 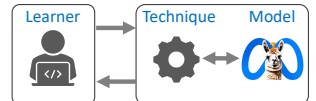 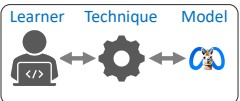

(a) Workflow using OpenAI's GPT-4.    (b) Workflow using open-access models.    (c) Hints-In-Browser.

Figure 1: Deployment workflows for generating programming feedback; see Section 1 for details.

metadata gets sent to external servers and further to closed-access corporate-run models [3, 11, 13]. While some of these concerns can be partially alleviated with alternative deployment workflows that leverage open-access models like Llama-3-70B [14, 15] (see Figure 1b), the underlying issues of cost (for hosting/serving models), time, and data privacy remain.

**Hints-In-Browser.** In this paper, we benchmark language models for programming feedback generation while considering different performance criteria important for real-world deployments, including quality, cost, time, and data privacy. To tackle the above-mentioned issues, we leverage recent advances in the new paradigm of in-browser inference that allow these models to be run directly in the end user's browser. In fact, we show how to directly integrate such an in-browser inference engine with modern browser-based programming environments [16, 17]. We refer to this new deployment workflow of in-browser feedback generation as *hints-in-browser* (see Figure 1c).

**Summary of Results.** Compared to existing workflows discussed earlier (see Figure 1), hints-in-browser provides clear benefits across two of our new performance criteria: (i) no running costs for an educator or learner; (ii) complete data privacy as the learner's data does not leave their machine. In fact, after an initial download and caching of the model inside the browser, the learner can continue to benefit from hints-in-browser even without internet connectivity. Given that in-browser inference engines are still in their infancy, they do put stringent requirements on feasible model sizes and consumer hardware. To further enhance and demonstrate the utility of hints-in-browser, we take the following steps:

- We develop a fine-tuning pipeline using GPT-4 generated synthetic data to boost the feedback quality of small models compatible with in-browser engines.[1] We showcase the efficacy of fine-tuned Llama-3-8B [14, 15] and Phi-3-3.8B [18] 4-bit quantized models using the WebLLM engine [19] on different Python programming datasets and highlight competitive quality surpassing GPT-3.5 [20].

- We build a web app to demonstrate and reliably measure the performance of hints-in-browser in terms of inference times and functionality across different hardware configurations.[2] We highlight that inference times are competitive with existing feedback generation workflows for compatible machines (e.g., a GPU-equipped consumer laptop, starting at the 1200 USD price range).

- We release the full implementation of our work along with the web app and datasets to facilitate further research and development on hints-in-browser.

## 2    Related Work

**Small Models and In-Browser Inference.** There have been increasing efforts to democratize generative AI technology. On the one hand, this has led to a surge of increasingly powerful small open-access models like Llama-3-8B [14, 15] and Phi-3-3.8B [18], thereby reducing the inference-time compute requirements. On the other hand, this has led to new in-browser inference engines such as WebLLM and ONNX Runtime that enable inference directly in browser [19, 21], without even requiring any installation as needed by end-user technologies like Ollama [22]. Recent works have also evaluated small open-access models for program repair, highlighting their potential in educational settings [23]. We contribute to this ecosystem by benchmarking feedback generation techniques across important real-world performance criteria.

**Deployment of Feedback Generation Techniques.** There has been an increasing interest in deploying feedback-generation techniques for learners/teachers in real-world settings. Recent research works have demonstrated pilot deployment of programming feedback techniques in classrooms based on OpenAI's models [10, 11, 24]. However, given the running costs involved, it

---

[1]Github repository: https://github.com/machine-teaching-group/neurips2024-hints-in-browser-finetuning.
[2]Github repository: https://github.com/machine-teaching-group/neurips2024-hints-in-browser-demo.

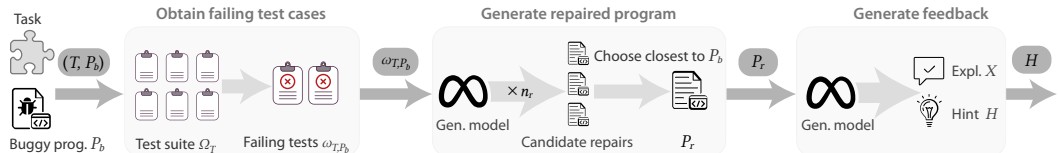

Figure 2: Illustration of different steps in the feedback generation technique (adapted from [5, 9]).

is infeasible for educators to provide large-scale free access to such technology. Beyond research, educational websites have also been integrating generative feedback techniques: Khanamigo by Khan Academy [25] and Q-Chat by Quizlet [26] are AI-powered systems based on OpenAI's GPT-4 models. However, these systems are typically subscription-based to cover the running costs. These recent developments and limitations motivate our work to study and benchmark new deployment workflows.

## 3 Programming Feedback and Performance Metrics

Next, we introduce programming feedback scenarios and technique along with performance metrics.

### 3.1 Programming Feedback Scenarios and Technique

**Hints Scenario.** Given a programming task $T$ and a learner's buggy program $P_b$, we aim to generate a natural language hint $H$ as feedback to aid the learner in resolving programming errors and making progress. The buggy program $P_b$ fails to pass at least one of the test cases in the task $T$'s test suite $\Omega_T$ and may contain various error types, including syntax and semantic errors. The objective is primarily to aid the learner with a concise hint without directly revealing the solution or required fixes [5, 9, 11].

**Program Repair Scenario.** Given a programming task $T$ and a learner's buggy program $P_b$, we aim to generate a repaired (fixed) program $P_r$. The objective is to make minimal edits to the buggy program $P_b$, and preserve context and style as much as possible. While this fixed program $P_r$ is not directly used to provide feedback to learners, it is typically used as an intermediate step when generating hints. For a given generative model, the quality of generated program repairs is a good proxy for the quality of generated hints, and can be assessed automatically [5].

**Feedback Generation Technique.** Figure 2 illustrates the feedback generation technique we consider in this paper adapted from existing techniques [5, 9]. We provide an overview of the key steps and hyperparameters, which inspire the fine-tuning pipeline in Section 4 and affect our results on inference time in Section 5. The starting point is a learner working on a programming task $T$ who has written a buggy program $P_b$ and requests a hint. The first step obtains a small set of failing test cases from the test suite $\Omega_T$, denoted as $\omega_{T,P_b}$. The second step generates a repaired program using the generative model by prompting it with $(T, P_b, \omega_{T,P_b})$. More specifically, we request the model to generate a candidate repaired program, and independently repeat this process $n_r$ times with temperature $t_r$. Then, from this set of $n_r$ programs, we choose the program that passes the test suite $\Omega_T$ and is closest to $P_b$ in terms of token-edit distance. This is based on the Levenshtein edit distance on strings obtained by tokenizing the programs, as done in previous work [9]. Henceforth, we refer to this repaired program as $P_r$; if no repaired program is found, we define $P_r$ to be empty. The next step uses all this information and makes a new query to the model asking to generate a concise hint $H$ along with an auxiliary detailed explanation $X$ as part of Chain-of-Thought reasoning [27]. Finally, the hint $H$ is provided to the learner as feedback. The prompts used are adapted from existing literature [9, 11] and are provided in our Github repository.

### 3.2 Performance Metrics

**Quality.** We use existing quality metrics for hints and repair as considered in [5, 9]. For hints, we assess the quality of generated hints along four binary attributes, which are computed based on expert ratings: (a) *HCorrect* captures whether the generated hint provides correct information for resolving issues in the buggy program; (b) *HInformative* captures whether the generated hint provides useful information to help the learner resolve bug(s); (c) *HConceal* captures that the information in the generated hint is not too detailed, so the learner would also have to reason about implementing the fixes; (d) *HComprehensible* captures whether the generated hint is easy to understand, presented in

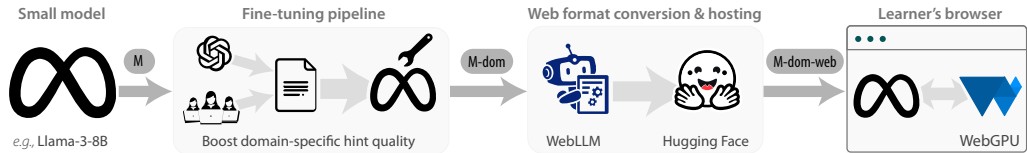

Figure 3: Illustration of the hints-in-browser deployment workflow; see Section 4 for details.

a readable format, and does not contain redundant information. Following the setup from existing literature [5, 9], we measure the overall quality of the generated hint by *HGood* that takes the value of $1$ (good quality) when all the four attributes are rated as $1$. For repair, we assess the quality of generated programs along two quality attributes: (a) *RPass* measures whether any of the $n_r$ candidate repaired programs passes the $T$'s test suite $\Omega_T$ (i.e., pass@$n_r$ rate [28]); (b) *REdit* captures the token-based edit distance between the repaired program and the buggy program (computed when a repaired program is found).

**Cost, Time, and Data Privacy.** We use a richer set of performance criteria important for real-world deployments, including cost, time, and data privacy. The *cost* metric here refers to the running monetary costs for an educator or learner for a single hint request. The *time* metric here refers to waiting times for a learner while the technique generates and serves hints. The *data privacy* metric here refers to privacy from the learner's perspective of how their data is sent to external services.

## 4 Hints-In-Browser Workflow and Fine-Tuning Pipeline

In this section, we provide an overview of the methodology, including the hints-in-browser deployment workflow and the fine-tuning pipeline.

**Hints-In-Browser Workflow.** Figure 3 provides an overview of the hints-in-browser deployment workflow and its interaction with a learner requesting hints on a programming task. The starting point is a small language model as a base model in 16-bit format; we will use Llama-3-8B [14, 15] or Phi-3-3.8B [18] throughout this paper. The next stage is to optionally fine-tune the base model to boost the feedback generation quality using the pipeline discussed below. Next, the model is converted to its web version, which also quantizes the model to a 4-bit format suitable for an in-browser inference engine. In this paper, we use the WebLLM engine [19], which makes use of recently added WebGPU capabilities in several modern browsers to enable fast in-browser inference. Next, the web version of the model and auxiliary inference files are hosted in a publicly accessible space such as on HuggingFace [29]. Finally, the hints-in-browser workflow is deployed and accessed by a learner in the form of a web app, which includes a programming environment along with the option to request hints (see Section 6). When the learner accesses the app for the first time, the model is downloaded and cached locally. On subsequent hint requests, our technique runs in the browser and utilizes WebLLM-powered inference to directly provide feedback to the learner. As discussed in Section 1, hints-in-browser provide direct benefits across the performance criteria of cost and data privacy.

**Fine-Tuning Pipeline to Boost Quality.** We use the smallest models from Llama-3 and Phi-3 families for hints-in-browser to keep inference times low. To boost the feedback quality of these small models, we develop a fine-tuning pipeline; see Figure 3. The training process involves standard supervised fine-tuning with LoRA [30] and uses GPT-4 generated synthetic training data. Next, we discuss the main steps involved in this synthetic training data generation process; see Figure 4a. The starting point is a set of programming tasks $\{T\}$ that are used for training; this set corresponds to a (sub)set of tasks in the domain that a learner will be working on. The first step is to obtain buggy programs $\{P_b\}$ for each task $T$ along with failing test cases $\omega_{T,P_b}$ (see Section 3.1 for this notation). These buggy programs can be real-world learner submissions or synthetic programs obtained by prompting GPT-4 – we will consider both settings in the experiments. When using GPT-4 to generate buggy programs, we prompt it for types of real-world mistakes (e.g., syntactical errors, incorrect loop range); see Figure 4b. The second step is to prompt the GPT-4 model with $(T, P_b, \omega_{T,P_b})$ and ask it to generate a repaired program $P_r$, a detailed explanation $X$, and a hint $H$ in a single prompt shown in Figure 4c; here, we adapt the prompts used in the feedback generation technique. From $(T, P_b, \omega_{T,P_b}, P_r, X, H)$, we assemble four types of training data instances as shown in Figure 4a. We provide further implementation details of our synthetic training data generation process in our Github repository.

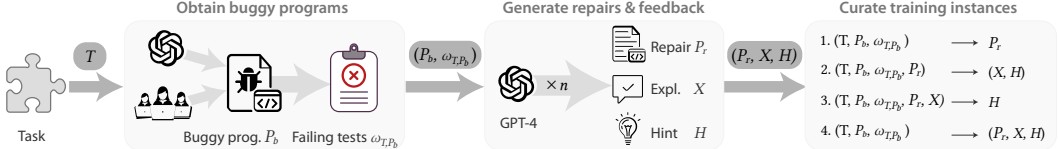

(a) Illustration of the synthetic training data generation process; see Section 4 for details.

| Synthetic Data: Buggy Programs |
| --- |
| You are working on a Python programming problem. Can you write buggy codes from a student's perspective who makes mistakes while solving the problem? Below, you are first provided the problem description and then the type of mistakes which the student makes. |
| Problem description: `{problem_description}` |
| Type of mistakes: `{type_of_mistakes_description}` |
| Can you generate 5 different buggy programs which are diverse in terms of code structure and style, while making the above mentioned type of mistakes? |

| Synthetic Data: Repairs and Feedback |
| --- |
| I'm working on a Python programming problem. The current program below is not working well. Can you help in fixing this program and giving feedback about the fixes? Below I first provide the problem description, failing test cases, and then the buggy program. |
| Problem description: `{problem_description}` |
| Failing test cases: `{failing_test_cases}` |
| Buggy program: `{buggy_program}` |
| (1) Can you fix the above buggy program while making minimal changes needed to fix the program? (2) Can you describe the bug(s) in this program and the required fixes? (3) Can you provide a concise hint about one bug in this program? Be socratic and friendly. |

(b) Prompt for synthetic data (buggy programs)  (c) Prompt for synthetic data (repairs and feedback)

Figure 4: Synthetic training data generation process along with *shortened version* of prompts used for generation. Details are skipped for brevity and full prompts are provided in our Github repository.

# 5 Evaluation: Fine-Tuning Experiments and Feedback Quality

In this section, we evaluate different model variants to assess the impact of web conversion and effectiveness of our fine-tuning pipeline in terms of feedback quality.

## 5.1 Programming Domains and Datasets

We consider three domains/datasets representing a variety of Python programming tasks; see Figure 5. We provide a high-level overview here and defer the full details to the our Github repository.

**BASICALGO [5, 9].** The first dataset, BASICALGO, has been introduced in the literature to benchmark generative models and human tutors in terms of feedback generation quality [5, 9]. It comprises 5 popular Python programming tasks that involve writing basic algorithms, namely GCD, FIBONACCI, DIVISORSDIV3, PALINDROME, and MERGESTRS. For each task, the dataset contains 5 buggy programs based on bugs in real-world submissions on the *geeksforgeeks.org* platform [31]. These buggy programs represent a variety of non-trivial bugs, including algorithmic mistakes, time/space complexity issues, and Python-specific errors like incorrect string indexing. We use these 25 buggy programs for evaluation. For the fine-tuning pipeline, we begin by generating synthetic buggy programs using GPT-4 in Step-1 of the synthetic training data generation process in Figure 4a. Then, for each buggy program in the training set, we use GPT-4 to obtain up to 5 independent tuples $(P_r, X, H)$; here, we filter invalid tuples, e.g., where $P_r$ is not a correct repair. Finally, we assemble 4 training instances per tuple, which leads to a total of 3304 instances as reported in Figure 5a.

**INTROPYNUS [32, 33].** The second dataset, INTROPYNUS, has been introduced in the literature to benchmark program repair techniques [32, 33]. It comprises 5 Python programming tasks that involve writing functions, namely DUPLICATEELIMINATION, SORTINGTUPLES, TOP-k ELEMENTS, SEQUENTIALSEARCH, and UNIQUEDATESMONTHS. For each task, the dataset contains (buggy and correct) submissions from 361 students from an introductory Python course taught at the National Univer-

| Properties | Domain:BASICALGO | Domain:INTROPYNUS | Domain:KARELALGO |
|---|---|---|---|
| Domain and concepts | Basic algorithms in Python | Basic functions in Python | Specialized linked lists |
| Type of programming tasks | Real-world | Real-world | Designed |
| Number of programming tasks | 5 | 5 | 7 |
| Avg. lines in solution programs | 7.4 | 8.4 | 11.3 |
| Anticipated difficulty level | Medium | Easy | Hard |
| Evaluation: Type of buggy programs | Real-world | Real-world | Designed |
| Evaluation: Number of buggy programs | 25 | 25 | 21 |
| Evaluation: Avg. lines in buggy programs | 11.8 | 11.0 | 10.3 |
| Fine-tuning: Type of buggy programs | GPT-4 generated | Real-world | GPT-4 generated |
| Fine-tuning: Number of buggy programs | 191 | 1758 | 301 |
| Fine-tuning: Number of training instances | 3304 | 30576 | 3676 |

(a) Summary of important properties across domains/datasets.

---

Given a `string s`, return 1 if `s` is a palindrome, otherwise return 0. Expected time complexity should be $O(\text{Length of } s)$ and expected auxiliary space should be $O(1)$. Example: `is_palindrome`("abba") $\rightarrow 1$.

---

(b) PALINDROME task from BASICALGO.

---

Write a function that takes in a `list lst` and returns a new `list` with all repeated occurrences of any element removed. Relative order of the elements should be preserved. Example: `remove_extras`([5, 2, 1, 2, 3]) $\rightarrow$ [5, 2, 1, 3].

---

(c) DUPLICATEELIMINATION task from INTROPYNUS.

---

Given a `linklist a`, check if it is a palindrome. This `linklist a` is palindrome if the values of its elements and elements in its reverse are the same. You are also given an additional `linklist b` that can be used to reverse `linklist a`, and then compare. Note that the initial pointers for `linklist a` and `linklist b` are at their first elements. Your function should return a boolean value `True` or `False`. Example: `is_palindrome`(a:[4, 5, 7, 5, 4], b:[0, 0, 0, 0, 0]) $\rightarrow$ `True`.

This domain has specialized linked-lists inspired by Karel environment. The possible operations on `linklist a` are:
`a.go_next()`, `a.go_prev()`, `a.get_value()`, `a.set_value()`, `a.has_next()`, `a.has_prev()`.

---

(d) PALINDROME task from KARELALGO.

Figure 5: Overview of domains and datasets used for evaluation, along with example tasks.

sity of Singapore. We select a small set of diverse buggy programs, 5 per task, to curate an evaluation set of 25 buggy programs. For the fine-tuning pipeline, we use the remaining buggy programs as real-world buggy programs in Step-1 of the synthetic training data generation process in Figure 4a. Then, for each buggy program in the training set, we use GPT-4 to obtain 5 independent tuples $(P_r, X, H)$. Finally, we assemble 4 training instances per tuple, which leads to a total of 30576 instances.

**KARELALGO.** The third dataset, KARELALGO, has been designed by us and involves writing Python programs in a new domain inspired by the Karel visual programming environment [34, 35]. Importantly, these tasks require using domain-specific operations that would have not been seen by the generative models during training. In short, we consider 1-D Karel-worlds which leads to solving linked-list variant of problems, with environment-specific operations (see Figure 5d). This dataset comprises 7 programming tasks that involve writing functions with environment-specific operations, namely MOVE, COPYENDTOALL, SUMOFLIST, NTHELEMENTENDOFLIST, COMMONDIVISOR, PALINDROME, and FIBONACCISERIESUPTON. For each task, the dataset contains 3 buggy programs that we designed to capture a mixture of environment-specific bugs and typical bugs in similar problems. We use these 21 buggy programs for evaluation. For the fine-tuning pipeline, we begin by generating synthetic buggy programs using GPT-4 in Step-1 of the synthetic training data generation process in Figure 4a and follow the same process as used for BASICALGO, which leads to a total of 3676 instances.

## 5.2 Models Studied and Experimental Details

**Models Studied.** Next, we summarize different models and variants used in the feedback generation technique. Unless specified, we use the same hyperparameters across all models given by the number of repairs $n_r = 10$, repair temperature $t_r = 0.7$, and hint temperature of $t_h = 0.1$.

Table 1: Results for feedback quality performance for technique with different models. Averaged results are reported as mean (stderr) and rounded off for brevity. See details in Section 5.3.

| Technique | Domain:BASICALGO | | | Domain:INTROPYNUS | | | Domain:KARELALGO | | | All Domains |
|---|---|---|---|---|---|---|---|---|---|---|
| $n_r = 10$ | HGood% | RPass% | REdit | HGood% | RPass% | REdit | HGood% | RPass% | REdit | HGood% |
| GPT-4 | 76 | 97 (1) | 45 (1) | 100 | 97 (1) | 15 (1) | 90 | 100 (0) | 30 (1) | 89 ( 7) |
| GPT-4o | 68 | 96 (0) | 29 (1) | 92 | 100 (0) | 14 (0) | 90 | 95 (0) | 20 (0) | 83 ( 8) |
| GPT-4o mini | 60 | 96 (0) | 44 (1) | 88 | 99 (1) | 16 (1) | 67 | 76 (3) | 31 (0) | 72 ( 8) |
| GPT-3.5 | 24 | 88 (0) | 39 (2) | 48 | 95 (3) | 14 (1) | 29 | 75 (3) | 13 (1) | 34 ( 7) |
| Llama-3-8B | 32 | 83 (1) | 44 (1) | 52 | 93 (1) | 19 (2) | 24 | 41 (3) | 19 (1) | 36 ( 8) |
| Llama-3-8B-dom | 64 | 83 (1) | 35 (2) | 96 | 100 (0) | 17 (0) | 71 | 90 (3) | 22 (0) | 77 (10) |
| Llama-3-8B-web | 16 | 67 (5) | 55 (3) | 36 | 89 (1) | 21 (2) | 10 | 40 (3) | 20 (1) | 21 ( 8) |
| Llama-3-8B-dom-web | 52 | 83 (1) | 40 (0) | 84 | 100 (0) | 21 (1) | 57 | 84 (2) | 23 (1) | 64 (10) |
| Phi-3-3.8B | 36 | 84 (4) | 43 (1) | 32 | 71 (5) | 29 (2) | 14 | 33 (3) | 39 (6) | 27 ( 7) |
| Phi-3-3.8B-dom | 48 | 73 (3) | 40 (1) | 68 | 100 (0) | 13 (0) | 38 | 67 (3) | 14 (1) | 51 ( 9) |
| Phi-3-3.8B-web | 4 | 55 (4) | 51 (5) | 16 | 71 (4) | 26 (3) | 5 | 29 (7) | 31 (8) | 8 ( 4) |
| Phi-3-3.8B-dom-web | 24 | 56 (2) | 43 (2) | 64 | 93 (1) | 12 (1) | 19 | 46 (4) | 18 (4) | 36 (14) |

Table 2: Results for feedback quality performance with a lower value of $n_r$ (total number of program repairs generated by the model as part of the hint generation process). Choosing a higher value can increase hint quality but also increase the inference time. See details in Sections 5.3 and 6.

| Technique | Domain:BASICALGO | | | Domain:INTROPYNUS | | | Domain:KARELALGO | | | All Domains |
|---|---|---|---|---|---|---|---|---|---|---|
| $n_r = 3$ | HGood% | RPass% | REdit | HGood% | RPass% | REdit | HGood% | RPass% | REdit | HGood% |
| GPT-4 | 68 | 97 (1) | 48 (1) | 96 | 91 (1) | 17 (2) | 86 | 97 (2) | 37 (0) | 83 ( 8) |
| GPT-4o mini | 56 | 92 (2) | 45 (3) | 88 | 96 (2) | 18 (1) | 52 | 59 (2) | 28 (2) | 65 (11) |
| Llama-3-8B-dom-web | 36 | 67 (5) | 46 (2) | 84 | 100 (0) | 25 (1) | 57 | 71 (7) | 26 (1) | 59 (14) |
| Phi-3-3.8B-dom-web | 24 | 36 (2) | 36 (6) | 60 | 87 (1) | 15 (0) | 14 | 37 (2) | 19 (3) | 33 (14) |

- **OpenAI's GPT family.** We consider models from OpenAI's GPT family, including GPT-4 (model=*gpt-4-turbo-2024-04-09*) [1, 20], GPT-4o (model=*gpt-4o-2024-08-06* ), GPT-4o mini (model=*gpt-4o-mini-2024-07-18*), and GPT-3.5 (model=*gpt-3.5-turbo-0125*) [20]. These models are popularly used in current deployments of programming feedback techniques [10, 11].

- **Base and fine-tuned models before web conversion.** As base models, we consider the smallest models from Llama-3 and Phi-3 families, namely Llama-3-8B (model=*Meta-Llama-3-8B-Instruct*) [14, 15] and Phi-3-3.8B (model=*Phi-3-mini-4k-instruct*) [18]. For each of the three domains, we use our fine-tuning pipeline to obtain a domain-specific model, referred to by the suffix *-dom*. We use the same fine-tuning hyperparameters throughout the model families and domains.

- **Base and fine-tuned models after web conversion.** For each of the base models and their fine-tuned variants, we perform web conversion using WebLLM libraries [19]. As part of this conversion, we quantize the model to a 4-bit format suitable for the WebLLM inference engine. We refer to a web version of a model with the suffix *-web*.

**Quality Assessment.** As discussed in Section 3.2, we use quality metrics from literature and report performance for the two scenarios of hint generation and program repair. For the program repair quality metrics, we use test suites to automatically compute correctness and edit tokens. For hint quality performance, we follow the procedure used in literature [5, 9] where we ask human evaluators to assess the quality of generated hints w.r.t. the quality attributes. To begin, we did a small-scale investigation to establish the reliability of the rating criteria, where two human evaluators (comprising one author and one external researcher) independently rated 90 generated feedback instances; we obtained a Cohen's kappa reliability value of $0.644$, which indicates *substantial agreement* between evaluators [36] and matches values observed in the literature [9]. Afterward, given the scale of annotations, we did one human annotation per generated feedback for the final evaluation. In our results, we use *HGood%* and *RPass%* to denote the percentage of evaluation examples that have *HGood* equal to 1 and *RPass* equal to 1, respectively.

| Technique | Domain:KARELALGO | | | Domain:KARELALGO:NEW | | |
|---|---|---|---|---|---|---|
| $n_r = 10$ | HGood% | RPass% | REdit | HGood% | RPass% | REdit |
| Llama-3-8B | 24 | 41 (3) | 19 (1) | 22 | 37 (10) | 18 ( 2) |
| Llama-3-8B-dom | 71 | 90 (3) | 22 (0) | 67 | 74 ( 7) | 33 ( 4) |
| Phi-3-3.8B | 14 | 33 (3) | 39 (6) | 0 | 7 ( 4) | 17 (10) |
| Phi-3-3.8B-dom | 38 | 67 (3) | 14 (1) | 33 | 48 ( 7) | 21 ( 2) |

| Dataset (%) | Domain:INTROPYNUS | | |
|---|---|---|---|
| $n_r = 10$ | minEdit | RPass% | HGood% |
| 0 | 82 (10) | 93 (1) | 52 |
| 10 | 24 ( 4) | 91 (1) | 60 |
| 50 | 10 ( 2) | 92 (0) | 68 |
| 100 | 6 ( 1) | 100 (0) | 96 |

(a) Results on new tasks in a domain.      (b) Results with varying dataset sizes.

Figure 6: (a) Results for KARELALGO and KARELALGO:NEW; (b) Results for Llama-3-8B-dom for INTROPYNUS when varying the size of the training dataset.

**Details About Fine-Tuning Experiments**. We use the same hyperparameters across all models and domains. For training, we use standard supervised fine-tuning with LoRA [30]. Full implementation details are available in our Github repository. We conducted both the training and inference of the open-access models on a cluster of machines with Nvidia A-100 GPUs.

## 5.3 Results for Quality Performance

For each model and domain, we report aggregated performance over $(T, P_r)$ instances in the evaluation set. For the program repair scenario, we compute performance from three independent runs and report averaged results as mean (stderr). For models with fine-tuning, a run refers to training a new model using our fine-tuning pipeline followed by inference. For the hints generation scenario, we report performance for only the first run, given the scale of annotations; here, we also report aggregate results across domains. Table 1 shows the results for quality performance and we summarize the main takeaways below.

**Effect of Fine-Tuning.** We first look at the effectiveness of the fine-tuning pipeline. The performance of base models (Llama-8B and Phi-3.8B) is substantially lower than GPT-4 and on par with GPT-3.5. After fine-tuning, the models' performance (Llama-8B-dom and Phi-3.8B-dom) surpasses GPT-3.5 on all domains; in particular, Llama-8B-dom's performance surpasses GPT-4o-mini on all domains and comes close to GPT-4 on INTROPYNUS.

**Effect of Web Conversion.** An important question is how web conversion and quantization from 16-bit to 4-bit representation affects the quality of feedback generation, as model sizes reduce by a factor of over 3.5 times. Table 1 showcases the performance of 4 models along with their web versions. When looking at aggregated results for all domains on hints quality, we see that Llama-3 models show a smaller drop in performance after web conversion, in contrast to drop for Phi-3 models.

**Effect of Number of Repairs.** An important hyperparameter is $n_r$, i.e., the total number of program repairs generated by the model as part of the hint generation process. Choosing a higher value can increase hint quality but also increase the inference time. Table 2 shows results for feedback quality performance with a lower value of $n_r = 3$, in contrast to $n_r = 10$ in Table 1. These results show that $n_r = 3$ could provide a reasonable quality-time trade-off, especially, on INTROPYNUS.

**Generalization for New Tasks in a Domain**. In Figure 6a, we report additional results on how the fine-tuned models perform on new tasks in the same domain that were unseen during training. Here, we consider the KARELALGO domain and introduce 3 new programming tasks in the domain, namely ARITHMETICSERIESUPTON, REVERSELIST and CONCATENATELISTS. We call this new evaluation set as KARELALGO:NEW that also includes 3 buggy programs for each of the new tasks. We see that the performance boost by fine-tuning is transferred to new problems in the same domain.

**Varying the Size of the Training Dataset**. In Figure 6b, we show additional results where we vary the percentage of the training dataset used for the INTROPYNUS domain. The column labeled *minEdit* measures the token edit distance between a buggy program in the evaluation set and its closest counterpart in the corresponding training set, reported as mean (stderr).

## 6 Evaluation: Web App And Inference Times

As noted earlier, in-browser inference engines are still in their infancy and put stringent requirements on consumer hardware and feasible model sizes. We build a web app to demonstrate and reliably measure the performance across different hardware configurations, and facilitate further research on hints-in-browser. Next, we provide an overview of the web app and then report on the results.

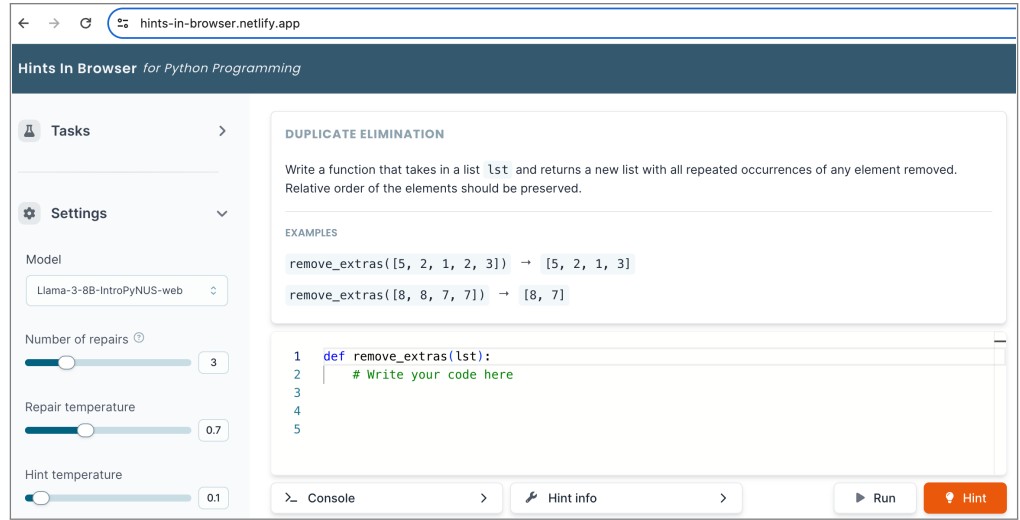

Figure 7: Screenshot of hints-in-browser app at `https://hints-in-browser.netlify.app/`.

**Web App.** The web app demonstrating the hints-in-browser workflow is accessible at `https://hints-in-browser.netlify.app/`; see a screenshot in Figure 7. This web app provides a Python programming environment for learners: they can select a programming task in the domain and write/execute programs. This programming environment uses Pyodide [17], a popular browser-based Python interpreter that allows for in-browser Python execution without the need to interact with a server. The feedback generation technique is exposed to learners as a hint button. The model is downloaded and cached locally when the learner accesses the web app for the first time. When a learner requests a hint in the web app, our technique runs on the browser using the WebLLM engine and provides hints to the learner directly in the app. After an initial download and caching of the model, a learner can continue to work on programming tasks and benefit from hints-in-browser without internet connectivity. We provide the full implementation in our Github repository.

**Results Across Hardware Configurations.** Next, we report results regarding the inference time during hint generation, model reloading time from the browser cache whenever the app is refreshed, and model downloading time initially.[3] In-browser inference engines require a GPU-equipped machine; hence, we only consider compatible hardware configurations accessible to us. Table 3 provides a comparison of different deployment workflows for feedback generation, as shown in Figure 1, while considering different hardware configurations. More concretely, we consider the INTROPYNUS domain, set $n_r = 3$, and report results averaged over five programming tasks. Our results highlight that the inference times for in-browser inferences are competitive with existing feedback generation workflows for compatible machines (e.g., a GPU-equipped consumer laptop starting at $1200$ USD); machines with more powerful GPUs report a drop in inference times to 20 seconds for the Phi-3-3.8B model. It is worth noting that we could achieve lower inference times by reducing the number of intermediate repairs and potentially by fine-tuning a base model to reduce the number of output tokens generated, e.g., fewer tokens generated as part of the chain-of-thought process. Table 3 further reports on the performance criteria of cost per hint request and privacy aspects of the learner's data.

**Results for Local Models Running Natively.** We further evaluated the latencies of a 4-bit quantized Llama-3-8B model running natively on an Apple M1 Pro Laptop, using the same evaluation setup as in Table 3 in two formats: GGUF [37] and MLX [38]. The GGUF format model executed using Ollama had an inference time of about 70 seconds, while the MLX format model (optimized for Apple silicon via the MLX framework) had an inference time of 35 seconds. In comparison, the web version of the model using the WebLLM in-browser inference engine had an inference time of about 70 seconds, thereby closely matching the inference time of the GGUF format model. Notably, in comparison to local models running natively, in-browser inference engines offer significant practical advantages, such as cross-platform compatibility through WebGPU, ease of access without additional setup, and enhanced security via a sandboxed environment.

---

[3]For the web models, typical initial download times on a 100 Mbps connection are around 6 mins for Llama-3-8B (4.2GB file) and 3 mins for Phi-3-3.8B (2GB file).

Table 3: Comparison of different deployment workflows for feedback generation, as shown in Figure 1. Results are reported for the INTROPYNUS domain and $n_r = 3$. Under the "Hardware Configuration", L: indicates laptops and D: indicates desktops. Under the "Performance", the "Inference (s)" and "Cost (USD)" columns report results per hint request, and the "Privacy" column highlights privacy aspects of the learner's data across different workflows. (*We estimated the energy cost of locally invoking the domain-specific web models on a Apple M1 Pro, i.e., L:M1 Pro and it was approximately $6.8 \times 10^{-5}$ USD.)

| Model | Hardware Configuration | | | Performance | | | |
|---|---|---|---|---|---|---|---|
| | Machine | Year | Price (USD) | Inference (s) | Reload (s) | Cost (USD) | Privacy |
| GPT-4 | n/a | n/a | n/a | 34 | 0 | $5.7 \times 10^{-2}$ | Fig 1a: External org |
| GPT-4o mini | n/a | n/a | n/a | 15 | 0 | $1.0 \times 10^{-3}$ | Fig 1a: External org |
| Llama-3-8B-dom | n/a | n/a | n/a | 31 | 0 | $2.1 \times 10^{-3}$ | Fig 1b: External server |
| Llama-3-8B-dom-web | L:M3 Max | 2024 | 3500 | 28 | 7 | $0^*$ | Fig 1c: User local |
| | L:M2 Pro | 2023 | 2200 | 56 | 9 | | |
| | L:M1 Pro | 2021 | 1600 | 71 | 12 | | |
| | L:M2 Air | 2022 | 1100 | 129 | 12 | | |
| | D:RTX 3080 Ti | 2021 | 2000 | 38 | 22 | | |
| | L:RTX 3060 | 2021 | 1200 | 48 | 25 | | |
| | L:RTX 2060 | 2021 | 850 | 624 | 30 | | |
| Phi-3-3.8B-dom-web | L:M3 Max | 2024 | 3500 | 20 | 4 | $0^*$ | Fig 1c: User local |
| | L:M2 Pro | 2023 | 2200 | 34 | 4 | | |
| | L:M1 Pro | 2021 | 1600 | 45 | 6 | | |
| | L:M2 Air | 2022 | 1100 | 54 | 10 | | |
| | D:RTX 3080 Ti | 2021 | 2000 | 36 | 11 | | |
| | L:RTX 3060 | 2021 | 1200 | 44 | 15 | | |
| | L:RTX 2060 | 2021 | 850 | 310 | 14 | | |

# 7 Concluding Discussions

We investigated language models for programming feedback generation and benchmarked their performance across several criteria crucial for real-world educational deployments. We showed how to leverage the new paradigm of in-browser inference that allows these models to run directly inside the browser and introduced *hints-in-browser* as a new deployment workflow. As a concrete instance of hints-in-browser, we showcased the efficacy of a fine-tuned Llama-3-8B and Phi-3-3.8B 4-bit quantized models using the WebLLM engine across different performance criteria.

**Limitations and Future Work.** Next, we discuss some limitations of our current work and ideas for tackling them in the future. First, we experimented with Llama-3-8B and Phi-3-3.8B as base models; it would be interesting to consider variants of Llama-3-8B and Phi-3-3.8B fine-tuned on programming datasets and even smaller models possibly obtained by distilling these base models. In particular, it would be useful to experiment with new small models from the Llama-3.2 family, which also have the potential to perform well on mobile and edge hardware [39]. Second, we leveraged WebLLM's in-browser inference engine; it would be useful to compare it with alternatives such as the ONNX Runtime engine. Third, we studied the scenario of programming feedback generation; it would be worth exploring in-browser language models for other domains. Last, it would be interesting to examine how this workflow performs in terms of sustainability. While we compared different workflows primarily based on quality and cost, it would also be useful to conduct a systematic evaluation based on energy usage.

**Broader Implications.** Our experimental results and the web app demonstrate the potential of in-browser models in powering next-generation technologies for programming education. While we do not foresee any negative societal impacts of our work, we note that in-browser inference engines are still in their infancy and have stringent requirements on feasible model sizes and consumer hardware. In the coming years, we expect rapid advances in this ecosystem, fueled by upcoming in-browser inference engines, increasing capabilities of small generative models, and growing compute resources in consumer hardware. The results in this paper and the implementation, along with the web app and datasets, contribute to this ecosystem and will facilitate further research on hints-in-browser.

## Acknowledgments and Disclosure of Funding

Funded/Co-funded by the European Union (ERC, TOPS, 101039090). Views and opinions expressed are however those of the author(s) only and do not necessarily reflect those of the European Union or the European Research Council. Neither the European Union nor the granting authority can be held responsible for them.

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
