# A Datasheet

We include the datasheet for both our curated benchmark and the synthetically generated dataset.

## A.1 Motivation

**For what purpose was the dataset created?**

The evaluation datasets were curated to benchmark language models for programming feedback generation across several performance criteria. The synthetic datasets were generated to improve the models' performance on generating program repairs and feedback. These datasets were used for fine-tuning with the aim to improve the feedback quality in introductory Python programming domains.

**Who created the dataset (e.g., which team, research group) and on behalf of which entity (e.g., company, institution, organization)?**

We have three different Python programming domains: BASICALGO, INTROPYNUS, and KARELALGO. Each domain has a training set and an evaluation set. For BASICALGO, the evaluation set was curated by the authors of [9, 5] and the training set was generated by us (the authors of this paper). For INTROPYNUS, the evaluation set and the starter buggy programs were curated by the authors of [32], and the training set was generated by us (the authors of this paper). For KARELALGO, the tasks, evaluation sets, and training sets were curated/generated by us (the authors of this paper).

**Who funded the creation of the dataset?**

Details about funding for this work are provided in the acknowledgements.

**Any other comments?**

None.

## A.2 Composition

**What do the instances that comprise the dataset represent (e.g., documents, photos, people, countries)?**

The dataset consists of Python programming tasks, buggy attempts on those tasks, a fixed version of each attempt, along with explanations and hints. Prompt templates are also included.

**How many instances are there in total (of each type, if appropriate)?**

BASICALGO:

Tasks: 5, Evaluation Program: 25, Training Instances: 3304

INTROPYNUS:

Tasks: 5, Evaluation Programs: 25, Training Instances: 30576

KARELALGO:

Tasks: 7, Evaluation Programs: 21, Training Instances: 3676

**Does the dataset contain all possible instances or is it a sample (not necessarily random) of instances from a larger set?**

It contains all possible instances.

**What data does each instance consist of?**

Each instance of a task consists of a task description, task constraints, and a set of test cases for the task. Each instance in the evaluation set consists of a pair of buggy and fixed Python programs for a specific task. Each instance in the training set consists of a buggy Python program, a fixed program, an explanation for the bugs, and a hint.

**Is there a label or target associated with each instance?**

Yes, all the training instances have input and output labels.

**Is any information missing from individual instances?**

Not to the best of our knowledge.

**Are relationships between individual instances made explicit (e.g., users' movie ratings, social network links)?**

N/A.

**Are there recommended data splits (e.g., training, development/validation, testing)?**

Yes, the data is naturally split into evaluation and training segments as discussed in the paper.

**Are there any errors, sources of noise, or redundancies in the dataset?**

The evaluation sets for the benchmark are created by experts and do not have errors to the best of our knowledge. The synthetically generated training data is validated automatically during creation. No part of the synthetic data has been checked manually for errors, sources of noise or redundancies which go beyond automated checks.

**Is the dataset self-contained, or does it link to or otherwise rely on external resources (e.g., websites, tweets, other datasets)?**

A part of the datasets we use are from previously published works [9, 5, 32].

**Does the dataset contain data that might be considered confidential (e.g., data that is protected by legal privilege or by doctor–patient confidentiality, data that includes the content of individuals' non-public communications)?**

The dataset for the KARELALGO domain that is specifically designed by us doesn't contain any confidential data.

**Does the dataset contain data that, if viewed directly, might be offensive, insulting, threatening, or might otherwise cause anxiety?**

No.

**Does the dataset identify any subpopulations (e.g., by age, gender)?**

No.

**Is it possible to identify individuals (i.e., one or more natural persons), either directly or indirectly (i.e., in combination with other data) from the dataset?**

No.

**Does the dataset contain data that might be considered sensitive in any way (e.g., data that reveals race or ethnic origins, sexual orientations, religious beliefs, political opinions or union memberships, or locations; financial or health data; biometric or genetic data; forms of government identification, such as social security numbers; criminal history)?**

No.

**Any other comments?**

None.

### A.3 Collection Process

**How was the data associated with each instance acquired?**

The tasks and evaluation datasets for the BASICALGO and the INTROPYNUS domains are based on their respective sources [9, 5, 32]. The tasks and the evaluation dataset for KARELALGO is designed by the authors of this paper. For all the domains, the synthetic training datasets were generated by the authors of this paper using OpenAI's GPT-4 model.

**What mechanisms or procedures were used to collect the data (e.g., hardware apparatuses or sensors, manual human curation, software programs, software APIs?**

The evaluation datasets were collected or curated manually, while the training datasets were generated using OpenAI's GPT-4 model.

**If the dataset is a sample from a larger set, what was the sampling strategy (e.g., deterministic, probabilistic with specific sampling probabilities)?**

N/A.

**Who was involved in the data collection process (e.g., students, crowdworkers, contractors) and how were they compensated (e.g., how much were crowdworkers paid)?**

Only the authors were involved in the data collection process.

**Over what timeframe was the data collected?**

The training datasets for all the domains were collected from March to May 2024. For BASICALGO, we also used a part of datasets previous published work in 2023 [5]; for INTROPYNUS, we also used a part of datasets from previous published work in 2019 [32].

**Were any ethical review processes conducted (e.g., by an institutional review board)?**

N/A.

**Did you collect the data from the individuals in question directly, or obtain it via third parties or other sources (e.g., websites)?**

N/A.

**Were the individuals in question notified about the data collection?**

N/A.

**Did the individuals in question consent to the collection and use of their data?**

N/A.

**If consent was obtained, were the consenting individuals provided with a mechanism to revoke their consent in the future or for certain uses?**

N/A.

**Has an analysis of the potential impact of the dataset and its use on data subjects (e.g., a data protection impact analysis) been conducted?**

N/A.

**Any other comments?**

None.

## A.4 Preprocessing/cleaning/labeling

**Was any preprocessing/cleaning/labeling of the data done (e.g., discretization or bucketing, tokenization, part-of-speech tagging, SIFT feature extraction, removal of instances, processing of missing values)?**

In our synthetic data generation process, we have automated checks to validate generated buggy programs and their repairs. We do not do any manual cleaning of the data beyond the automated checks.

**Was the "raw" data saved in addition to the preprocessed/cleaned/labeled data (e.g., to support unanticipated future uses)?**

N/A.

**Is the software that was used to preprocess/clean/label the data available?**

The full implementation is provided as part of our Github repository.

**Any other comments?**

None.

### A.5 Uses

**Has the dataset been used for any tasks already?**

The data from BASICALGO and INTROPYNUS was previously used in their intended ways by the authors of their papers [9, 5, 32]. The synthetic dataset was only used in this paper.

**Is there a repository that links to any or all papers or systems that use the dataset?**

N/A.

**What (other) tasks could the dataset be used for?**

The dataset can be used for evaluation and fine-tuning generative models for programming feedback generation. In future, the datasets could potentially be used by researchers in other suitable ways as well.

**Is there anything about the composition of the dataset or the way it was collected and preprocessed/cleaned/labeled that might impact future uses?**

Not to the best of our knowledge.

**Are there tasks for which the dataset should not be used?**

Training on this data does not guarantee generalization to other domains.

**Any other comments?**

None.

### A.6 Distribution

**Will the dataset be distributed to third parties outside of the entity (e.g., company, institution, organization) on behalf of which the dataset was created?**

Our Github repository contains the full evaluation and training dataset for the KARELALGO domain.

**How will the dataset will be distributed (e.g., tarball on website, API, GitHub)?**

The KARELALGO domain dataset will be available on our Github repository.

**When will the dataset be distributed?**

The KARELALGO domain dataset will be available on our Github repository at the time of publication.

**Will the dataset be distributed under a copyright or other intellectual property (IP) license, and/or under applicable terms of use (ToU)?**

The KARELALGO domain dataset will be available on our Github repository under CC license.

**Have any third parties imposed IP-based or other restrictions on the data associated with the instances?**

No.

**Do any export controls or other regulatory restrictions apply to the dataset or to individual instances?**

No.

**Any other comments?**

None.

### A.7 Maintenance

**Who will be supporting/hosting/maintaining the dataset?**

The authors of the paper will provide the required maintenance to the dataset.

**How can the owner/curator/manager of the dataset be contacted (e.g., email address)?**

Yes, the authors can be contacted by email for any queries related to the datasets.

**Is there an erratum?**

Not at this point of time.

**Will the dataset be updated (e.g., to correct labeling errors, add new instances, delete instances)?**

We may update the dataset to correct any labeling errors or improve its quality. For any such updates, a versioning system will be used to keep track of versions.

**If the dataset relates to people, are there applicable limits on the retention of the data associated with the instances (e.g., were the individuals in question told that their data would be retained for a fixed period of time and then deleted)?**

N/A.

**Will older versions of the dataset continue to be supported/hosted/maintained?**

In case new data will be released, a versioning system will be used to keep track of versions.

**If others want to extend/augment/build on/contribute to the dataset, is there a mechanism for them to do so?**

We plan to use GitHub for maintaining both the source code and datasets. Others who are interested in extending or contributing to this work can contact authors.

**Any other comments?**

None.