# OpenReview forum: "Hints-In-Browser: Benchmarking Language Models for Programming Feedback Generation"
_NeurIPS.cc/2024/Datasets_and_Benchmarks_Track — NeurIPS 2024 Track Datasets and Benchmarks Poster_

### Official Review · Reviewer_7NvE · 2024-07-17
**This paper provides a rigorous benchmark and implementation of LLMs for in-browser student feedback**

**Rating:** 9
**Confidence:** 4

**Review:**

See summary.

**Strengths:**

- The code is made available.
- The paper promises to make a more privacy sensitive environment for students.
- The paper provides a rigorous benchmark of multiple LLMs and different logical variants, such as web-compiled ones
- The paper provides a rigorous evaluation across different metrics, such as cost (of laptop and inference), time, privacy, and helpfulness of feedback.

**Additional Feedback:**

None.

**Clarity:**

The paper is extremely well written, easily to follow. Some very minor remarks.

- Some tables are referred to as "Figure 3" (should be "Table 1").
- Sometimes the method is referred to as "The Technique". Is this common in your field? I wasn't aware of this methodology, and would maybe instead refer to as the "method" or "teaching intervention" ?
- How was the annotation done in more detail by the humans? Were there instructions? Was this done by the authors? Was this always the same person or multiple?

**Correctness:**

The evaluation seems rigorous, providing multiple runs and standard deviations.

- I wasn't sure why 25 buggy programs are used for evaluation if more are available.
- Could it be the case that there are very-close to duplicates in the buggy programs (e.g. one program contains an extra "space" compared to another), leading to near duplicates in the finetuning and evaluation set? Can this be avoided? Otherwise, the evaluation might be overly optimistic.
- What is the rationale behind sometimes using "real world" "GPT-4" generated or "designed" buggy programs? Would it be possible to report results for multiple settings? Especially in some settings there seem to be enough real-world buggy programs available.
- How can be made sure the Karel visual programming environment was not in the GPT-4 training data?

**Documentation:**

I did not check the source code in detail. There seems to be sufficient detail to reproduce the work.

**Ethics:**

None.

**Opportunities For Improvement:**

There are some very minor issues.
- The evaluation metrics are a bit unclear. How it was explained in the paper, I assumed they are in [0,1], but they seem to be in [0,100] if I look at the results (line 120).
- One clear other alternative that is overlooked or not mentioned in the paper, is to have a server with a GPU running the inferences for a whole class. This can be a more cost efficient option, especially it can be unlikely for students to have strong GPU's in all countries.

**Relation To Prior Work:**

Yes, detailed in section 2.

**Summary And Contributions:**

This paper provides a benchmark of multiple LLMs and evaluates their usage for providing hints in the browser of a student making programming exercises. The tool is opensourced, and designed in such a way that the LLM runs entirely in the browser, improving privacy. Multiple LLMs are evaluated for this purpose and compared; in particular, LLMs are finetuned or not, and some are compressed (low precision weights) so that they can run on laptops with smaller GPUs. Prompts are designed to solicit the right kind of hints from the LLMs, and feedback by the LLMs are evaluated rigorously in a human evaluation. The methods are compared accross various different types of costs. E.g. GPT-4 is found to outperform some methods in terms of quality, but in terms of privacy and cost not. This provides a comprehensive benchmark accross multiple different criteria.

---

> ### Author Rebuttal · Authors · 2024-08-16
>
> **Response to Reviewer 7NvE (Part 1)**
>
> Thank you for carefully reviewing our paper! We greatly appreciate your feedback. Please see below our responses to your comments.
>
> -----
> **1. The evaluation metrics are a bit unclear. How it was explained in the paper, I assumed they are in [0,1], but they seem to be in [0,100] if I look at the results (line 120).**
>
> Thanks for pointing out this issue. We will update the paper to make this consistent. Indeed, for a given feedback instance, the evaluation for HGood is binary. In Figure 6 and Figure 7, we reported HGood in terms of the percentage of evaluation examples in which HGood is rated as 1 for the generated feedback.
>
> -----
> **2. One clear other alternative that is overlooked or not mentioned in the paper, is to have a server with a GPU running the inferences for a whole class.**
>
> The deployment workflow shown in Figure 1b is considered to capture the alternative as suggested by the reviewer. In the table below, we provide additional results where we also compare this deployment workflow with the other workflows analyzed in the paper.
>
> _Table: Comparison of different deployment workflows for feedback generation (based on Figure 1). Time and costs are shown for the IntroPyNUS domain with $n_r=3$._
>
> | Model | Workflow | Hardware Configuration | Privacy | Inference (s) | Reload (s) | Cost per hint (cents) |
> | -------- | ------- | -------  | -------  | -------  | -------  | -------  |
> | Llama-3-8B-dom-web | Hints-in-browser (Fig 1c) | Apple M1 Pro, 2021, 1600 USD | User local | 71 | 12 | 0* |
> | Phi-3-3.8B-dom-web | Hints-in-browser   (Fig 1c) | Apple M1 Pro, 2021, 1600 USD | User local | 45 | 6 | 0* |
> | Llama-3-70B-dom | Self Deployed  (Fig 1b)  | NA | External server | 95 | 0 | 1.5 |
> | gpt-4-turbo | OpenAI  (Fig 1a) | NA | External organization | 34 | 0 | 5.6 |
> | gpt-4o | OpenAI (Fig 1a) | NA | External organization | 12 | 0 | 2.5 |
>
>
> *There is additional energy cost involved in running the model locally, we will add this discussion in the conclusions section.
>
> -----
> **3. I wasn't sure why 25 buggy programs are used for evaluation if more are available.**
> - The human cost to annotate the feedback for HGood is quite significant. It takes a person roughly 1 minute to annotate one feedback instance (i.e., about 30 mins for 25 feedback instances). In our experiments, we had 25 feedback instances across 3 different domains and 14 different model/$n_r$ combinations. This corresponds to about 21 hours of human effort in annotations. This process is hard to scale and hence we scoped our evaluation to 25 programs per domain or approximately 75 programs in total. We would like to point out that the number of programs used in our work are of the same order of magnitude as used in related works on feedback generation that rely on human evaluation.
> - The evaluation for RPass and REdit can be done on a larger scale. In Figure 10 (Appendix D in supplementary material), we check RPass on a larger scale to pick a checkpoint post training. The same thing can be done for evaluation by creating a separate evaluation set for RPass and REdit.
>
> -----
> **4. Could it be the case that there are very-close to duplicates in the buggy programs (e.g. one program contains an extra "space" compared to another), leading to near duplicates in the finetuning and evaluation set?**
> - In Figure 11 (Appendix D in supplementary material), we also show generalization of the benefits of our fine-tuning to new tasks in a domain for Domain:KarelAlgo.
> - In the table below, we show additional results where we vary the percentage of the dataset used. Here, we also show a column _Distance_ (reported as mean and standard error) indicating the token edit distance between a buggy program in the evaluation set and its closest counterpart in the corresponding training set.
>
> _Table: Results for feedback quality performance for Llama-3-8B-dom on the IntroPyNUS domain with varying dataset sizes. Here $n_r=10$._
>
> | Training set used (in %) | _Distance_ | RPass | HGood |
> | -------- | ------- | -------  | -------  |
> | 0 | 82 (10) | 92 | 52 |
> | 10 | 24 (4) | 96 | 60 |
> | 50 | 10 (2) | 96 | 68 |
> | 100 | 6 (1) | 100 | 96 |
>
> - In general, we would like to note that if good representative data is available, or if the synthetic data generation technique works well, we expect real life buggy programs to be close to programs in the training set. This is one of the advantages of aligning a smaller model to a specific programming domain.
>
> -----
> (the response is continued in Part 2)

---

> ### Author Rebuttal · Authors · 2024-08-16
>
> **Response to Reviewer 7NvE (Part 2)**
>
> (continuation of the response from Part 1)
>
> -----
> **5. What is the rationale behind sometimes using "real world" "GPT-4" generated or "designed" buggy programs?**
> - In all the domains, by default we use the real-world data when available; however, we rely on synthetic data when we don't have access to real-world data. Figure 5a provides more details on the type of data that was available to us and how we used it for each domain. As an example, consider Domain:BasicAlgo: For this domain, we used an existing benchmark that provides 25 real-world buggy programs for evaluation; however, we do not have access to large-scale students’ buggy programs for fine-tuning and hence we rely on synthetically generated training data.
> - By considering these varied domains, we want to account for different real world settings where real data may not always be available.
>
> -----
> **6. How can be made sure the Karel visual programming environment was not in the GPT-4 training data?**
>
> The Domain:KarelAlgo considered in the paper has been introduced specifically for the experimental evaluation of our work. In particular, the domain comprises tasks with new specialized operations (shown in Figure 5d) that are adapted from the standard Karel. To the best of our knowledge, the publicly available tasks in the Karel domain do not use these specialized domain-specific operations.
>
> -----
> **7. Sometimes the method is referred to as "The Technique". Is this common in your field?**
>
> Thank you for pointing this out. We will make the terminology more clear in the updated paper. For instance, in Figure 1, we will make it clear that “technique” just refers to the method used for generating programming feedback.
>
> -----
> **8. How was the annotation done in more detail by the humans?**
>
> We briefly discussed this aspect in Section 5.2. Given the scale of the annotations, we did one human annotation per generated feedback for the final evaluation (here, one of the authors annotated the generated feedback instances). Before these final annotations, we did a small-scale investigation to establish the reliability of the rating criteria, where two human evaluators (comprising one author and one external researcher) independently rated 90 generated feedback instances. From this investigation, we got the Cohen’s kappa reliability value of 0.644, which indicates substantial agreement between evaluators and closely matches the values observed in literature.  Overall, the human annotation procedure in this paper closely follows the procedure considered in related literature.
>
> -----
> We hope that our responses can address your concerns. If you have any other comments or feedback, please let us know!

---

> ### Author Response · Authors · 2024-08-27
>
> Dear Reviewer 7NvE,
>
> We hope that our responses have addressed your concerns and questions. Given that the discussion period is ending soon, we would like to check if you have any further feedback about the paper.
>
> Sincerely,
>
> Authors

---

> > ### Comment · Reviewer_7NvE · 2024-08-30
> > **I am very satisfied**
> >
> > Thanks for your responses. I am very satisfied and will retain my high score.
> >
> > Best,
> > Reviewer

---

### Official Review · Reviewer_Hsuw · 2024-07-24

**Rating:** 6
**Confidence:** 3

**Review:**

The paper focuses on the application of in-browser inference. Therefore, the paper compares small models (rather than very large models) that are deployable to the inference, which sounds reasonable and inference. The paper also constructs new benchmark datasets for fine-tuning models to better hints generation, which should be useful for future research.

One potential improvement is documentation. It would be beneficial for users if there were detailed instructions about integrating customized/users' application-specific datasets into the proposed pipeline.

**Strengths:**

- Having a good user interface improves the applicability in real-world situations.

- Open-sourcing datasets is beneficial for the research community.

- Benchmarking only small models that are deployable to the in-browser inference is an interesting perspective. Also, the comparison of computational time is also useful from practical viewpoints.

**Additional Feedback:**

NA.

**Clarity:**

- Yes, the paper is clearly written.

**Correctness:**

- It is reasonable to compare the performance and runtime of small models as they are deployable to the in-browser inference framework.

- (While I am not an expert on this domain, ) the datasets seem to be constructed well.

**Documentation:**

- The details of the publicized datasets are provided in the paper.

- It would be beneficial if there were some documentations about the way of using customized dataset and fine-tuning users' own language model.

**Ethics:**

NA.

**Limitations:**

See opportunities for improvement.

**Opportunities For Improvement:**

- Because this paper presents an implementation pipeline, it would be useful to organize documentation with detailed explanations on how to use customized/application-specific data and users' own language model in the pipeline.

**Relation To Prior Work:**

- It is clearly written that the proposed framework puts emphasis on the use of in-browser inference for improved applicability.

- It is also clear that the publicized datasets are the first cost-free open-source dataset.

**Summary And Contributions:**

This paper presents a new pipeline for generating hints in programming using large language models. The key feature of the proposed pipeline is to enable simulation using in-browser inference, which allows users to meet data privacy without cost concerns. The paper also provides three benchmark datasets collected from real-world applications.

---

> ### Author Rebuttal · Authors · 2024-08-16
>
> **Response to Reviewer Hsuw**
>
> Thank you for carefully reviewing our paper! We greatly appreciate your feedback. Please see below our responses to your comments.
>
> -----
> **1. Because this paper presents an implementation pipeline, it would be useful to organize documentation with detailed explanations on how to use customized/application-specific data and users' own language model in the pipeline.**
>
> Thank you for the suggestion! Indeed, our pipeline can be used with any given language model and can be easily adapted to different programming domains. We will update the documentation with further details on how to apply the pipeline with new custom models or new programming domains.
>
> -----
> We hope that our responses can address your concerns. We have also provided implementation details and results in the supplementary material. If you have any other comments or feedback, please let us know!

---

> ### Author Response · Authors · 2024-08-27
>
> Dear Reviewer Hsuw,
>
> We hope that our responses have addressed your concerns and questions. Given that the discussion period is ending soon, we would like to check if you have any further feedback about the paper.
>
> Sincerely,
>
> Authors

---

> ### Comment · Reviewer_Hsuw · 2024-08-27
>
> Thank you for the responses. It's great that the authors plan to provide detailed documentation. I would maintain the positive evaluation as in the initial review.

---

### Official Review · Reviewer_5xx1 · 2024-07-25
**Paper review**

**Rating:** 6
**Confidence:** 4

**Review:**

The paper tackles an interesting and relevant problem in a novel manner, by incorporating code-related feedback with LLMs to the browser runtime via WebLLM. While this is a novel setting and potentially relevant to many downstream tasks beyond programming feedback, it lacks rigorous evaluation criteria in terms of the task at hand and context in terms of the computational runtime (please see following sections for more details). Additionally, the provided cost model does not have sufficiently motivate this serving scheme, especially under variable pricing, and while the concept of onloading computation may be conducive to the service provider, the end user may not be incentivised towards dedicating local resources. Last, the privacy extensions of local execution is important and should be included more prominently in the discussion.

**Strengths:**

* The paper does propose a novel setting for programming feedback generation by incorporating a computational setting that aligns with the way many platforms are served. The development of a web application to do so is an nice contribution.
* The use of various hardware resources and the associated latency and cost analysis is very welcome and refreshing.
* The impact of quantization to the quality aspects of the downstream task is an important dimension to capture.

**Additional Feedback:**

* The authors have only analysed the tractability of their models on desktop hardware. However, in the realm of education, many people nowadays leverage mobile-hardware in their devices, such as tablets. It would be nice to incorporate some discussion on mobile/edge hardware.
* What is the impact of running these models over WebLLM compared to natively?
* Another potential dimension to explore is personalizing the feedback to different cohorts of users, such as to beginners vs. advanced users.
* The paper seems to be exposing many generation hyperparameter values to the user, such as temperature, number of repairs, etc. An deeper analysis on the effect of these values and ideally an automated tuning would go a long way towards a better experience.
* I am curious why the authors have started from a non code-specific model, such as CodeGemma or Llama Code.

**Clarity:**

The paper is generally well written and easy to understand. It could be slightly refined by incorporating an algorithmic representation of the webapp's operation and training's workflow.

Some additional comments:
* "Python-specific errors like the mutability of lists": Maybe immutability of strings?
* It would be beneficial to the reader if the paper provided some actual examples of each task in §3.1.
* Tables should be captioned as such.

**Correctness:**

* The authors do not specify the experimental setup in terms of browser and framework version they have used to achieve the quoted performance.
* While the paper quantifies the cost of purchasing the hardware in Figure 9, it should also put it in context of the cloud-based cost of running these models as well as the OpenAI-based pricing.
* Some analysis on the size of the generated datasets as a function of downstream performance is missing from the manuscript.

**Documentation:**

The code is in good state and enables reproducibility by sufficient documentation and inclusion of the generated datasets. However, it is missing licensing information.

**Ethics:**

The proposed paradigm onloads computation to the consumer's device, which can increase cost for the user (at a micro scale) and can have negative impact on sustainability (at a macro scale). These dimensions should be discussed in my opinion in the context of broader impact.

**Limitations:**

* The authors have only experimented with 4-bit quantized architectures and LoRA fine-tuning. Whether this is the best setup for the downstream task has not been clear. Moreover, it would be nice to put it in the context of larger open-weight models, such as Llama-70B for example.
* The usage of GPT-4 for fine-tuning annotations potentially imposes certain restrictions in terms of downstream adoption (non-commercial licensing) and can leak user-data to a 3rd party. The authors should incorporate privacy in their discussion.
* The tasks only entail the Python language.
* Cost-wise:
    * The cost analysis only discusses the initial purchase cost and not the true ownership cost, which includes the energy consumption of the device.
    * The paper assumes a constant cost from using LLM-services in the cloud, which has not been the case over the past year. As various algorithmic, architectural and hardware optimisations get introduced, the total cost gradually becomes smaller. This dimension has not been sufficiently covered.

**Opportunities For Improvement:**

* The nature of the performance metrics is rather abstract and qualitatively defined, which leaves much room for subjectivity. Nevertheless, the identified dimensions seem very relevant to the downstream task.
* The inference and reload latencies from Figure 9 still look quite elevated and potentially bad for user Quality of Experience.
* It would be very interesting to measure the energy consumption for running these models locally in the browser, both from a cost and sustainability perpective.

**Relation To Prior Work:**

The authors could further incorporate the following work in their related work section:

* Majeed Kazemitabaar, Runlong Ye, Xiaoning Wang, Austin Zachary Henley, Paul Denny, Michelle Craig, and Tovi Grossman. 2024. CodeAid: Evaluating a Classroom Deployment of an LLM-based Programming Assistant that Balances Student and Educator Needs. In Proceedings of the CHI Conference on Human Factors in Computing Systems (CHI '24). Association for Computing Machinery, New York, NY, USA, Article 650, 1–20. https://doi.org/10.1145/3613904.3642773

**Summary And Contributions:**

This paper benchmarks language models for the task of generating programming feedback for coding tasks directly in the premises of a browser. To accomplish this, the authors create a workflow that leverages GPT-4 synthetic data to fine-tune smaller quantised models (llama-3, phi-3) on different Python programming feedback tasks. The runtime is based on WebLLM over a browser, aiming at a realistic web-application setting.

---

> ### Author Rebuttal · Authors · 2024-08-16
>
> **Response to Reviewer 5xx1 (Part 1)**
>
> Thank you for carefully reviewing our paper! We greatly appreciate your feedback. Please see below our responses to your comments.
>
> -----
> **1. The nature of the performance metrics is rather abstract and qualitatively defined, which leaves much room for subjectivity. Nevertheless, the identified dimensions seem very relevant to the downstream task.**
>
> We would like to clarify that the performance metrics used in the paper are taken from previous literature as cited in Section 3.2 of the paper. HGood [Phung et al. 2024], RPass or pass@nr rate [Chen at al. 2021], and REdit [Koutcheme 2023] have all been used for evaluation of code and feedback generated by language models.
>
> [Phung et al. 2024] Phung et al. “Automating Human Tutor-Style Programming Feedback: Leveraging GPT-4 Tutor Model for Hint Generation and GPT-3.5 Student Model for Hint Validation”. LAK 2024.
>
> [Chen at al. 2021] Chen et al. “Evaluating Large Language Models Trained on Code”. CoRR, abs/2107.03374, 2021.
>
> [Koutcheme, 2023] Koutcheme. “Training Language Models for Programming Feedback using Automated Repair Tools". AIED 2023.
>
> -----
> **2. The inference and reload latencies from Figure 9 still look quite elevated and potentially bad for user Quality of Experience.**
> - The primary goal of these experiments is to benchmark the performance of current state-of-the-art technology. As a point of comparison, when using the same feedback generation technique, we have also shown the inference times for GPT-4 in Figure 12 (Appendix E, supplementary material).
> - Furthermore, in the table below, we also include additional results which show that we could get lower latencies by setting  $n_r=0$ and removing chain-of-thought in the generation process (though with a drop in the feedback quality).
>
> _Table: Inference latencies for Apple M1 Pro on IntroPyNUS._
>
> | Model | HGood | RPass | Inference (s) |
> | -------- | ------- | -------  | -------  |
> | Llama-3-8B-dom-web ($n_r=3$)  | 84 | 100 | 71 |
> | Llama-3-8B-dom-web ($n_r=0$)  | 84 | NA | 33 |
> | Llama-3-8B-dom-web ($n_r=0$) (without CoT)  | 76 | NA | 11 |
> | GPT-4 ($n_r=3$)  | 96 | 92 | 34 |
> | GPT-4 ($n_r=0$)  | 92 | NA | 14 |
> | GPT-4 ($n_r=0$) (without CoT) | 72 | NA | 4 |
>
> -----
> **3. It would be very interesting to measure the energy consumption for running these models locally in the browser, both from a cost and sustainability perspective.**
>
> We provide a rough estimate for the additional energy consumption by locally monitoring the device’s energy consumption. As an example, we do it for Apple M1 Pro. For this, we use an open source tool (https://github.com/SAP/power-monitoring-tool-for-macos). We use the web-app with, and without hints and calculate the differential in power usage.
>
> - Additional average energy consumption while using these models locally on Apple M1 Pro is 19.68W. This value will vary between different hardware configurations.
> - Given that the average time taken to get feedback on this laptop is 71 seconds (for IntroPyNUS, $n_r=3$), the energy used for one request is 0.00038304 kWh.
> - The cost for this comes out to  0.00682 cents based on the average cost in the US (https://www.bls.gov/regions/midwest/data/averageenergyprices_selectedareas_table.htm).
>
> Apart from just the cost, It will be interesting to look at using LLMs locally from a sustainability perspective.  We will include this discussion in the updated paper.
>
> -----
> **4. While the paper quantifies the cost of purchasing the hardware in Figure 9, it should also put it in context of the cloud-based cost of running these models as well as the OpenAI-based pricing...  it would be nice to put it in the context of larger open-weight models, such as Llama-70B for example**
>
> - In the table below, we provide time and costs associated with feedback generation among different settings including OpenAI pricing.
> - Separately, we also did some additional experiments where we deployed a larger open-weight model (Llama-3-70B) on a GPU enabled server. This workflow corresponds to Figure 1b as shown in the paper. Please refer to the table below for additional results.
>
> _Table: Comparison of different deployment workflows for feedback generation (based on Figure 1). Time and costs are shown for the IntroPyNUS domain with $n_r=3$._
>
> | Model | Workflow | Hardware Configuration | Privacy | Inference (s) | Reload (s) | Cost per hint (cents) |
> | -------- | ------- | -------  | -------  | -------  | -------  | -------  |
> | Llama-3-8B-dom-web | Hints-in-browser (Fig 1c) | Apple M1 Pro, 2021, 1600 USD | User local | 71 | 12 | 0* |
> | Phi-3-3.8B-dom-web | Hints-in-browser   (Fig 1c) | Apple M1 Pro, 2021, 1600 USD | User local | 45 | 6 | 0* |
> | Llama-3-70B-dom | Self Deployed  (Fig 1b)  | NA | External server | 95 | 0 | 1.5 |
> | gpt-4-turbo | OpenAI  (Fig 1a) | NA | External organization | 34 | 0 | 5.6 |
> | gpt-4o | OpenAI (Fig 1a) | NA | External organization | 12 | 0 | 2.5 |
>
> *There is additional energy cost involved in running the model locally, we will add this discussion in the conclusions section.
>
> -----
> (the response is continued in Part 2)

---

> ### Author Rebuttal · Authors · 2024-08-16
>
> **Response to Reviewer 5xx1 (Part 2)**
>
> (continuation of the response from Part 1)
>
> -----
> **5. As various algorithmic, architectural and hardware optimisations get introduced, the total cost gradually becomes smaller. This dimension has not been sufficiently covered.**
> - We acknowledge that these costs are reducing over time. In the paper, we benchmarked the current best possible scenarios with local in-browser models and OpenAI models. In our updated paper, we will be including an analysis with respect to the new GPT-4o and GPT-4o-mini models.
> - We would like to emphasize that one of the significant advantages of using in-browser models is enhanced privacy.
>
> -----
> **6. The authors have only experimented with 4-bit quantized architectures and LoRA fine-tuning.**
>
> In the paper, we use the WebLLM engine by MLC, which does not support quantization larger than 4 bits. A recent work by [Dettmers et al. 2023] has shown that 4-bit quantization is optimal with respect to total model bits and performance. To allow the models to run on edge devices with reasonable memory requirements, 4 bit quantization seems to provide a good trade-off in terms of memory requirement.
>
> [Dettmers et al. 2023] Dettmers et al. “The case for 4-bit precision: k-bit inference scaling laws”. ICML 2023.
>
> -----
> **7. The usage of GPT-4 for fine-tuning annotations potentially imposes certain restrictions in terms of downstream adoption (non-commercial licensing) and can leak user-data to a 3rd party. The authors should incorporate privacy in their discussion.**
>
> Thank you for the suggestion, we will definitely incorporate privacy-related discussion more prominently in our updated paper. As future work, it would also be interesting to use an open-access model like Llama-3.1-405B, or Mistral Large to train smaller models for specific domains.
>
> -----
> **8. The tasks only entail the Python language.**
> - We chose the Python language as it has been more extensively studied in recent literature for feedback generation and code generation using language models. While we focused on Python, we included a variety of domains with different data availability settings (as shown in Figure 5a) within Python to test the applicability of these models in real life programming settings.
> - The overall pipeline and methodology of our work is quite general and can be adapted for feedback generation for other programming domains in future.
>
> -----
> **9. The authors do not specify the experimental setup in terms of browser and framework version they have used to achieve the quoted performance.**
> - For all of our experiments, we use the Chrome browser by Google, on both Windows and Mac OS. The version of the browser is post the April 2024 update in all cases.
> - For the WebLLM engine by MLC, we use version 0.2.40. The versions used for all the libraries and frameworks have been included in the appropriate requirement files in the code provided in the supplementary material.
>
> -----
> **10. Some analysis on the size of the generated datasets as a function of downstream performance is missing from the manuscript.**
>
> We include new results for this analysis in the table below.
>
> _Table: Results for feedback quality performance for Llama-3-8B-dom on the IntroPyNUS domain with varying dataset sizes. Here $n_r=10$._
>
> | Training set used (in %) | RPass | HGood |
> | -------- | -------  | -------  |
> | 0 | 92 | 52 |
> | 10 | 96 | 60 |
> | 50 | 96 | 68 |
> | 100 | 100 | 96 |
>
> -----
> (the response is continued in Part 3)

---

> ### Author Rebuttal · Authors · 2024-08-16
>
> **Response to Reviewer 5xx1 (Part 3)**
>
> (continuation of the response from Part 2)
>
> -----
> **11. What is the impact of running these models over WebLLM compared to natively?**
>
> In the table below, we include additional inference times where we show the performance of a 4-bit quantized Llama-3-8B model running natively on a M1 Pro Laptop. Here we run the model natively in two different formats - _gguf_ and _mlx_. The _-gguf_ model is run using Ollama, a popular cross-platform software to run language models on edge devices. The _-mlx_ model uses the MLX framework by Apple ML Research which is optimized for Apple silicon, and hence shows the best possible native performance. The optimum method to run the model natively will vary between different hardware configurations.
>
> _Table: Results for inference time for Llama-3-8B-dom running on Apple M1 Pro. The results are for the IntroPyNUS domain with $n_r=3$._
>
> | Model | Inference (s) |
> | -------- | -------  |
> | Llama-3-8B-dom -web | 71 |
> | Llama-3-8B-dom _-gguf_ | 69 |
> | Llama-3-8B-dom _-mlx_ | 35 |
>
> Beyond this runtime, there are several practical benefits of running the these models in-browser include:
> - Cross platform compatibility: WebLLM uses WebGPU which is meant to be supported across all platforms. This way the deployment does not differ across different platforms.
> - Ease of access: No additional setup is required from the user. It can be added directly to already existing in-browser programming platforms which provide no-setup access for new learners.
> - Sandboxed environment: From a student perspective, downloading external software can pose a security risk. Browsers provide a sandboxed environment, which can offer better security by limiting the access that the webpage has to the student’s system.
>
> We would include this discussion and analysis in the updated paper.
>
> -----
> **12. Many people nowadays leverage mobile-hardware in their devices, such as tablets. It would be nice to incorporate some discussion on mobile/edge hardware.**
>
> Thank you very much for the suggestion. We will include additional discussion on this in the updated paper.
>
> —--
> (the response is continued in Part 4)

---

> ### Author Rebuttal · Authors · 2024-08-16
>
> **Response to Reviewer 5xx1 (Part 4)**
>
> (continuation of the response from Part 3)
>
> -----
> **13. The paper seems to be exposing many generation hyperparameter values to the user, such as temperature, number of repairs, etc. An deeper analysis on the effect of these values and ideally an automated tuning would go a long way towards a better experience.**
> - We picked the repair temperature of 0.7 as it is the default temperature for most open-weight models, and we kept this uniform across all models.
> - In the table below, we provide additional results on how the performance varies across domains based on variation in $n_r$.
>
> _Table: Results for feedback quality performance across different values of $n_r$_
>
> | Technique |        | BasicAlgo |       |       |            | IntroPyNUS |       |       |            | KarelAlgo |       |       |            | All Domains |
> |-----------|--------|-----------|-------|-------|------------|---------|-------|-------|------------|------------|-------|-------|------------|-------------|
> |           | $n_r$ | HGood    | RPass | REdit |            | HGood   | RPass | REdit |            | HGood      | RPass | REdit |            | HGood       |
> | GPT-4     |   10     | 76        | 97 (1)    | 45 (1)  |            | 100         | 97 (1)    | 15 (1)    |            | 90         | 100 (0)  | 30 (1)  |            | 89 (7)    |
> | GPT-3.5   |     10   | 24        | 88 (0)   | 39 (2)   |            | 48          | 95 (3)   | 14 (1)   |            | 29         | 75 (3)    | 13 (1)  |            | 34 (7)        |
> | Llama-3-8B-dom-web | 10 | 52      | 83 (1)    | 40 (0)   |            | 84          | 100 (0)  | 21 (1)    |        | 57         | 84 (2)   | 23 (1)   |      | 64 (10)       |
> | Phi-3-3.8B-dom-web | 10 | 24      | 56 (2)    | 43 (2)   |            | 64          | 93 (1)    | 12 (1)  |            | 19 | 46 (4) | 18 (4) |  | 36 (14) |
> | GPT-4     |   3     | 68        | 97 (1)    | 48 (1)  |            | 96         | 91 (1)    | 17 (2)    |            | 86         | 97 (2)  | 37 (0)  |            | 83 (8)    |
> | GPT-3.5   |     3   | 20        | 75 (1)   | 40 (2)   |            | 44          | 91 (1)   | 16 (2)   |            | 33         | 67 (5)    | 13 (2)  |            | 32 (7)        |
> | Llama-3-8B-dom-web | 3 | 36     | 67 (5)    | 46 (2)   |            | 84          | 100 (0)  | 25 (1)    |        | 57         | 71 (7)   | 26 (1)   |      | 59 (14)       |
> | Phi-3-3.8B-dom-web | 3 | 24      | 36 (2)    | 36 (6)   |            | 60          | 87 (1)    | 15 (0)  |            | 14 | 37 (2) | 19 (3) |  | 33 (14) |
> | GPT-4     |   1     | 68        | 77 (3)    | 51 (1)  |            | 92         | 84 (2)    | 22 (1)    |            | 76         | 71 (7)  | 34 (5)  |            | 79 (7)    |
> | GPT-3.5   |     1   | 20        | 56 (4)   | 42 (2)   |            | 44          | 83 (3)   | 17 (1)   |            | 29         | 51 (6)    | 12 (2)  |            | 31 (7)        |
> | Llama-3-8B-dom-web | 1 | 36     | 45 (4)    | 60 (0)   |            | 88          | 92 (5)  | 27 (0)    |        | 43         | 56 (7)   | 24 (2)   |      | 56 (16)       |
> | Phi-3-3.8B-dom-web | 1 | 12      | 15 (3)    | 48 (14)   |            | 56          | 63 (1)    | 19 (3)  |            | 14 | 22 (3) | 21 (3) |  | 27 (14) |
> | GPT-4     |   0     | 60        |  NA    | NA  |            | 92         | NA    | NA    |            | 62         | NA | NA |            | 71 (10)    |
> | GPT-3.5   |     0   | 20        |   NA | NA   |            | 44          |  NA    | NA  |            | 24        |  NA    | NA  |             | 29 (7)        |
> | Llama-3-8B-dom-web | 0 |   32   |  NA    | NA  |           | 84           |  NA    | NA  |       | 24         |  NA    | NA  |     | 47 (19)       |
> | Phi-3-3.8B-dom-web | 0 |  12    |  NA    | NA  |           | 52          |  NA    | NA  |         | 14  |  NA    | NA  |  | 26 (13) |
>
> -----
> **14. I am curious why the authors have started from a non code-specific model, such as CodeGemma or Llama Code.**
>
> We picked the best general models in each size range. Among 8B models, Llama-3 is the most widely used. Also for coding, Llama-3-8B-Instruct outperforms CodeGemma and CodeLlama (based on Llama-2; note that Llama-3 does not have a code-specific version yet). Among 4B models, Phi-3-Mini-Instruct performs really well.
>
> [Liu et al. 2023] Liu et al. “Is your code generated by ChatGPT really correct? Rigorous evaluation of large language models for code generation”. NeurIPS 2023. https://evalplus.github.io/leaderboard.html
>
> -----
> **15. Another potential dimension to explore is personalizing the feedback to different cohorts of users, such as to beginners vs. advanced users.**
>
> Thank you for the suggestion. In future, we can look into specifically fine-tuning a model to help certain cohorts of students, along with fine-tuning them for a domain.
>
> -----
> **16. Clarify, additional comments, and relation to prior work.**
>
> Thanks for the additional feedback. We will carefully incorporate this in the final version of the paper. More concretely, we will update the paper as follows:
> - As suggested, we will incorporate an algorithmic representation of the webapp's operation and training's workflow.
> - We will update Section 3.1 with examples for clarity.
> - We will improve the captions/labels and fix typos wherever needed.
> - As suggested, we will update the related work section by adding discussion on CodeAid [Kazemitabaar et al., 2024].
>
> -----
> We hope that our responses can address your concerns and are helpful for improving your rating. Based on your feedback, we have provided additional details, empirical results, and analysis in the responses. If you have any other comments or feedback, please let us know!

---

> ### Author Response · Authors · 2024-08-27
>
> Dear Reviewer 5xx1,
>
> We hope that our responses have addressed your concerns and questions. Given that the discussion period is ending soon, we would like to check if you have any further feedback about the paper.
>
> Sincerely,
>
> Authors

---

> > ### Comment · Reviewer_5xx1 · 2024-08-29
> > **Response to the rebuttal**
> >
> > I would like to thank the authors for their extensive rebuttal and clarifications, as well as additional experiments. I have now revised my score.

---

> > > ### Author Response · Authors · 2024-08-30
> > >
> > > We sincerely thank the reviewer for their constructive feedback and engagement during the discussion period! We will incorporate the reviewer's feedback and our responses in the final paper. We appreciate the reviewer's input in helping us improve the paper. Thank you!

---

### Official Review · Reviewer_AZYp · 2024-08-17
**Review for "Hints-In-Browser: Benchmarking Language Models for Programming Feedback Generation"**

**Rating:** 7
**Confidence:** 3

**Review:**

1. The paper is thoroughly researched, offering a comprehensive
benchmarking of language models specifically designed for in-browser inference.
The methodology is robust, with a strong emphasis on the practical aspects of
implementing these models in educational environments.

2. The paper is well-written, with clearly organized sections that
effectively guide the reader through the research problem, methodology, and
findings. The figures and tables are helpful, effectively complementing the text.

3. The idea of utilizing in-browser inference engines for generating
programming feedback is creative and tackles important challenges in the field,
such as cost, scalability, and data privacy. The fine-tuning pipeline for smaller models is another novel contribution that enhances the practical usability of this
method.

4. This work is highly important for the educational technology
community, especially for those involved in automated feedback systems. The
approach has the potential to broaden access to high-quality programming
education, making it more scalable and cost-effective.

**Strengths:**

1. The use of in-browser inference engines for real-
time feedback generation is a novel and practical solution to existing
challenges in educational technology.

2. The paper provides a thorough
evaluation of various models across multiple performance criteria, offering
valuable insights into the trade-offs involved.

3. The development of a web app to demonstrate
the feasibility of the approach is a strong practical contribution that adds
significant value to the research.

4. The paper lays a solid foundation for future
research in this area, with clear suggestions for further exploration.

**Additional Feedback:**

The paper makes a strong contribution to the field of educational technology and
sets the stage for further research in this area. The practical implications of the
work are significant, and the paper is likely to have a positive impact on the
development of scalable, cost-effective, and privacy-conscious educational tools.
Further research could expand on this foundation by exploring additional models
and optimizing for a broader range of hardware configurations.

**Clarity:**

The paper is well written and clearly communicates its contributions. The
methodology is well-explained, and the results are presented in a logical and easy-
to-understand manner.

**Correctness:**

The claims made in the submission are correct, and the experimental design is
appropriate and performed correctly. The benchmark is constructed soundly, and
the evaluation methods are robust.

**Documentation:**

There is sufficient detail to support reproducibility, including descriptions of the
datasets, models, and web app. The paper includes all necessary information for
others to replicate the experiments and build upon the work.

**Ethics:**

There are no significant ethical concerns with this submission. The authors have
considered data privacy in their approach, which is a key strength of the in-
browser inference method.

**Limitations:**

The authors have adequately addressed the limitations of their work, including
the dependency on relatively powerful GPUs for in-browser inference and the
early stage of development of in-browser inference engines. However, further
exploration of smaller models and optimization for a broader range of hardware
would improve the approach's applicability.

**Opportunities For Improvement:**

1. The paper could explore a wider range of
models, including smaller or more specialized ones, to provide a broader
perspective on the potential of in-browser inference.

2. Further work could focus on
optimizing models for use on lower-end hardware, which would enhance
the accessibility of this approach.

3. Discussing the generalizability of
the web technologies used, such as WebLLM and WebGPU, across
different platforms would strengthen the paper.

**Relation To Prior Work:**

The paper clearly discusses how it differs from previous contributions,
particularly in its focus on in-browser inference and the specific challenges of
deploying programming feedback systems in educational settings.

**Summary And Contributions:**

Summary:

This paper examines the generation of programming feedback directly within the
browser, aiming to reduce costs and enhance data privacy. The authors evaluate
multiple models on aspects such as quality, cost, time, and privacy, and they fine-
tune smaller models using data generated by GPT-4. They also developed a web
application to demonstrate this approach, which proves to be competitive with
server-based methods.

Contributions:

1. This paper allows language models to run
directly in browsers, reducing costs and enhancing privacy.

2. This work enhances smaller models using GPT-4-
generated data for in-browser use, and advances scalable, cost-effective, and
privacy-conscious educational tools.

3. This paper evaluates models on feedback quality, cost,
time, and privacy.

---

> ### Author Rebuttal · Authors · 2024-08-17
>
> **Response to Reviewer AZYp**
>
> Thank you for carefully reviewing our paper! We greatly appreciate your feedback. Please see below our responses to your comments.
>
> -----
> **1. The paper could explore a wider range of models, including smaller or more specialized ones, to provide a broader perspective on the potential of in-browser inference.**
>
> Thank you for the suggestion. As future work, it would indeed be very interesting to consider smaller models possibly obtained by distilling Llama-3-8B and Phi-3-3.8B for programming domains. In the updated paper, we will also incorporate a related work in the discussion that has benchmarked smaller language models (with 164 million to 3 billion parameters) for educational program repair [Koutcheme et al., 2024].
>
> [Koutcheme et al. 2024] Koutcheme et al. “Benchmarking Educational Program Repair”. NeurIPS’23 Workshop on Generative AI for Education (GAIED).
>
> -----
> **2. Further work could focus on optimizing models for use on lower-end hardware, which would enhance the accessibility of this approach.**
>
> Optimizing models for specific lower-end hardware is indeed a very interesting direction of future work. Here, the performance could be improved across various aspects: (i) using more efficient inference engines; (ii) leveraging smaller domain-specialized models; (iii) fine-tuning the models to reduce the extra output tokens (e.g, related to chain-of-thought) needed for generating feedback.
>
> For instance, regarding aspect (i), it would be interesting to compare WebLLM’s in-browser inference engine with recently introduced alternatives such as the ONNX Runtime engine.
>
> -----
> **3. Discussing the generalizability of the web technologies used, such as WebLLM and WebGPU, across different platforms would strengthen the paper.**
>
> Thank you for the suggestion. We will be including additional discussion on advantages of using WebLLM and WebGPU in general, such as:
> - Cross platform compatibility: WebLLM uses WebGPU which is meant to be supported across all platforms. This way the deployment does not differ across different platforms.
> - Ease of access: No additional setup is required from the user. It can be added directly to already existing in-browser programming platforms which provide no-setup access for new learners.
> - Sandboxed environment: From a student perspective, downloading external software can pose a security risk. Browsers provide a sandboxed environment, which can offer better security by limiting the access that the webpage has to the student’s system.
>
> -----
> We hope that our responses can address your concerns. If you have any other comments or feedback, please let us know!

---

> ### Author Response · Authors · 2024-08-27
>
> Dear Reviewer AZYp,
>
> We hope that our responses have addressed your concerns and questions. Given that the discussion period is ending soon, we would like to check if you have any further feedback about the paper.
>
> Sincerely,
>
> Authors

---

### Author Rebuttal · Authors · 2024-08-16

**Common Response to Reviewers**

Dear Reviewers,

Thank you for carefully reviewing our paper and sharing feedback! For each reviewer separately, below we have provided detailed responses to address the questions and raised concerns. We will incorporate these responses in the updated version and truly believe that the paper will further improve based on your feedback.

Sincerely,

Authors

---

### Decision · Program_Chairs · 2024-09-26

**Decision:**

Accept (Poster)

**Comment:**

This paper benchmarks language models for programming feedback generations. All reviewers provide positive scores after rebuttal. Ac read all of them and agrees with the reviewers' recognition of this paper. AC hopes the authors can add the rebuttal in the final versions.